# High-confidence 3D template matching for cryo-electron tomography

Sergio Cruz-León [1], Tomáš Majtner [2], Patrick C. Hoffmann [2], Jan Philipp Kreysing [2,3], Sebastian Kehl [4], Maarten W. Tuijtel [2], Stefan L. Schaefer [1], Katharina Geißler [2,3], Martin Beck [2,5] ✉, Beata Turoňová [2] ✉ & Gerhard Hummer [1,6] ✉

Visual proteomics attempts to build atlases of the molecular content of cells but the automated annotation of cryo electron tomograms remains challenging. Template matching (TM) and methods based on machine learning detect structural signatures of macromolecules. However, their applicability remains limited in terms of both the abundance and size of the molecular targets. Here we show that the performance of TM is greatly improved by using template-specific search parameter optimization and by including higher-resolution information. We establish a TM pipeline with systematically tuned parameters for the automated, objective and comprehensive identification of structures with confidence 10 to 100-fold above the noise level. We demonstrate high-fidelity and high-confidence localizations of nuclear pore complexes, vaults, ribosomes, proteasomes, fatty acid synthases, lipid membranes and microtubules, and individual subunits inside crowded eukaryotic cells. We provide software tools for the generic implementation of our method that is broadly applicable towards realizing visual proteomics.

Cryo-electron tomography (CryoET) images the cellular environment in situ without labels and with fully preserved context[1,2]. Recent advances in hardware and acquisition techniques have enabled CryoET to routinely image, with high throughput, cell volumes in their native state and obtain structures of abundant macromolecular complexes with near molecular resolution[3–5]. However, lacking a uniform established method, the localization of particles in the tomograms remains highly customized, specific to each target, at best semi-automatic, and relying on strong manual input (such as the definition of geometric surface for large pleomorphic assemblies)[6–10] or extensive, often manual corrections for the false positives in an initial automated assignment[3,4,11]. The confident identification of a sufficient number of particles for a challenging target such as the nuclear pore complexes (NPC) can thus take months or years of manual annotation of literally hundreds of tomograms[12,13]. An automated, general, and reliable localization method would bring us closer to realizing the promise of visual proteomics[14–17] to build molecularly detailed representations of complex cellular landscapes from CryoET data.

Reliable assignment of molecular identities in tomograms is challenging due to both the biological context and the specifics of CryoET processing. Cells are crowded environments, and the proteins within them are structurally heterogeneous and vary widely in size and abundance. The physical limitations of the acquisition procedure further complicate particle localization[18–20]. In CryoET, the electron dose is limited to prevent sample radiation damage, which results in a low signal-to-noise ratio in the acquired tilt series. The maximum sample tilt of about ±60 degrees results in incomplete angular sampling known as the missing wedge problem in the three-dimensional (3D)

[1]Department of Theoretical Biophysics, Max Planck Institute of Biophysics, Max-von-Laue-Str. 3, 60438 Frankfurt am Main, Germany. [2]Department of Molecular Sociology, Max Planck Institute of Biophysics, Max-von-Laue-Str. 3, 60438 Frankfurt am Main, Germany. [3]IMPRS on Cellular Biophysics, Max-von-Laue-Str. 3, 60438 Frankfurt am Main, Germany. [4]Max Planck Computing and Data Facility, Gießenbachstraße 2, 85748 Garching, Germany. [5]Institute of Biochemistry, Goethe University Frankfurt, 60438 Frankfurt am Main, Germany. [6]Institute of Biophysics, Goethe University Frankfurt, 60438 Frankfurt am Main, Germany. ✉e-mail: martin.beck@biophys.mpg.de; beata.turonova@biophys.mpg.de; gerhard.hummer@biophys.mpg.de

reconstruction. In addition, the electron micrographs are conventionally captured out of focus. To recover the high-resolution information, it is thus necessary to accurately determine the defocus and correct the contrast transfer function (CTF)[20]. Visual proteomics needs to overcome these challenges for the reliable assignment of molecular identities to noisy 3D images of highly complex cellular volumes.

Manual tomogram annotation is still widely used despite being labor intensive, intrinsically subjective, and incomplete[21]. Template-based computational approaches[22] use known objects (templates) and compare them with the data by calculating a similarity metric (usually a constrained cross-correlation)[15,16,23–25]. In contrast, template-free methods iterate to cluster particles and determine patterns without imposing any structure[26,27]. However, their accuracy and efficiency need improvement[22]. Deep-learning algorithms, including classification and semantic segmentation, have been applied to CryoET[22,28–30]. Recently, implementations such as DeepFinder[29], DeePiCt[30], and TomoTwin[31] have shown promising results in segmenting tomograms and identifying the positions of common macromolecular complexes. However, these methods require extensive annotations for training and are less effective in detecting low-abundance particles[22], so far limiting their use to detect ribosomes and similarly sized particles. Furthermore, they determine only positions, and further processing is needed to determine particle orientations.

Template matching (TM)[15,16,23–25] is typically used with low-resolution templates of the macromolecular complex of interest on down-sampled tomograms to reduce computational cost and avoid template bias. Large numbers of false positive hits are removed either manually, thereby lowering the objectivity of the approach, or through a multistep classification procedure, which is computationally expensive and can fail if the number of particles is small. In addition, the data down-sampling limits the ability to localize smaller or weak-signal particles[32]. In theory, the ability of TM to localize the particles with high confidence should be connected to the quality of the template and how well it resembles the actual data. However, in practice, it has not been objectively shown how TM depends on the type of template and parameters such as voxel size, masks, resolution filtering, and the number of orientations.

In this study, we establish a high-confidence TM pipeline and combine it with CryoET imaging for visual proteomics of eukaryotic cells. We show that the performance of TM not only depends on the size, but also on its experimental origin and shape, and more importantly on the angular increment in orientational sampling in a template-specific manner. Furthermore, tomogram voxel size (magnification), filtering, and resolution have to be considered as optimization parameters. We demonstrate the power of optimized TM to localize nuclear pore complexes (NPCs), vault proteins, ribosomes, proteasomes, microtubules, and lipid membranes, inside a single dataset. We establish that TM can identify low-abundance and low-density complexes with high fidelity, as exemplified by the identification of ribosome-loaded vaults. We show that TM quantitatively captures conformers and subunits and provide recommendations for users to optimally set template-dependent search parameters and a parameter estimation software tool.

## Results

### High-confidence template matching for in situ macromolecule localization

We comprehensively tested our TM pipeline on tomograms of *Dictyostelium discoideum* and exemplifying tomograms from *Schizosaccharomyces pombe* and human tissue culture (Hek293) cells obtained from lamellae milled with cryo-focused ion beam microscopes[4] (see Fig. 1 for the workflow and "Methods" section for details on data acquisition). Starting from a library of the best available templates for a series of candidate features, we performed TM of each

template in a tomogram independently and assigned particle identities to the points with high constrained cross-correlation (CC). The locations and orientations of the assigned peaks permit the visualization and analysis of the spatial interactions of the features. We used a total of 21 templates, on 3 different species (Table 1) at different voxel sizes and with multiple search parameters including the number of orientations and filters (see "Methods" section for details). Templates in the library were obtained from different sources including subtomogram averaging (STA), homology modeling, the protein data bank (PDB)[33], the electron microscopy data bank (EMDB)[34], and molecular dynamics simulations (see "Methods" section for details).

We used the STOPGAP[35,36] software framework, and re-implemented it as a GPU-accelerated version (https://gitlab.mpcdf.mpg.de/bturo/gapstop_tm), to calculate the actual cross-correlation between templates and tomograms, maximizing the cross-correlation of the template according to its orientation and positions. This framework takes into account the missing wedge, angular tilt step, defocus, and electron dose (see "Methods" section and ref. 36 for details). For each template, with optimized search parameters (see next section), peaks several standard deviations above noise appear in the z-score map. High-confidence peaks correspond to the position where the center of the template is placed to best reproduce the data from the tomograms.

Figure 1 summarizes the TM procedure. We used a library that includes templates for the NPC[37], the 80S ribosome[4], and the nuclear envelope obtained by STA from tomograms of *D. discoideum*. For the proteasome[38] and microtubule[39], we used the previously reported human structures (PDB-id: 6rgq [https://www.rcsb.org/structure/6RGQ] (human 20S proteasome structure), PDB-id: 3jar [https://www.rcsb.org/structure/3JAR] (microtubule structure), respectively). For the vault, we created a density map starting from an atomic model generated by homology modeling. With each of the templates, we performed TM, initially at 4-binned data with a voxel size of 8.704 Å and then also at higher resolution (2-binned 4.352 Å/voxel and unbinned 2.176 Å/voxel). By progressing hierarchically to higher resolution, we aimed to capitalize on the high signal content of the data collected with the latest-generation hardware.

We transformed the cross-correlation volumes into z-score maps by subtracting the average and then dividing by the standard deviation (σ), both calculated for each template across the entire map. We use z-scores unless otherwise stated, as they quantify peak heights relative to the background in a particular tomogram. In the z-score representation, a peak at the center of the NPC is typically ~10 standard deviations (σ) above the map noise, while the vault and the ribosome have peaks with z-score values of ~30 and ~40, respectively (Fig. 1). For isolated objects such as the vault or ribosome, the peaks appear insular and sharp, while membrane or microtubules show elongated and continuous peaks consistent with the extended and repetitive character of the objects. Remarkably, TM identifies also low-density and low-abundance particles with high fidelity (Fig. 1). Automatic and semi-automatic particle detection algorithms have been widely tested for high-contrast and abundant macromolecular complexes in tomograms (e.g., ribosomes). However, fundamental macromolecular complexes such as the NPC or vault, which are scarce (2–3 copies per tomogram) and have low density, are particularly challenging. With optimal parameters, TM results in strong peaks for both macromolecular complexes (Fig. 1) and finds all positions identifiable by expert inspection. This finding is important in two ways: firstly, these complexes are fundamental for our understanding of cellular function, and secondly, given their low abundance, harnessing all the particles is key for visual proteomics analysis.

### Assessment of parameters that impact on the performance of TM

The success is dependent on the accurate tuning of various parameters, but clear guidelines on how to adjust those are missing. We analyzed

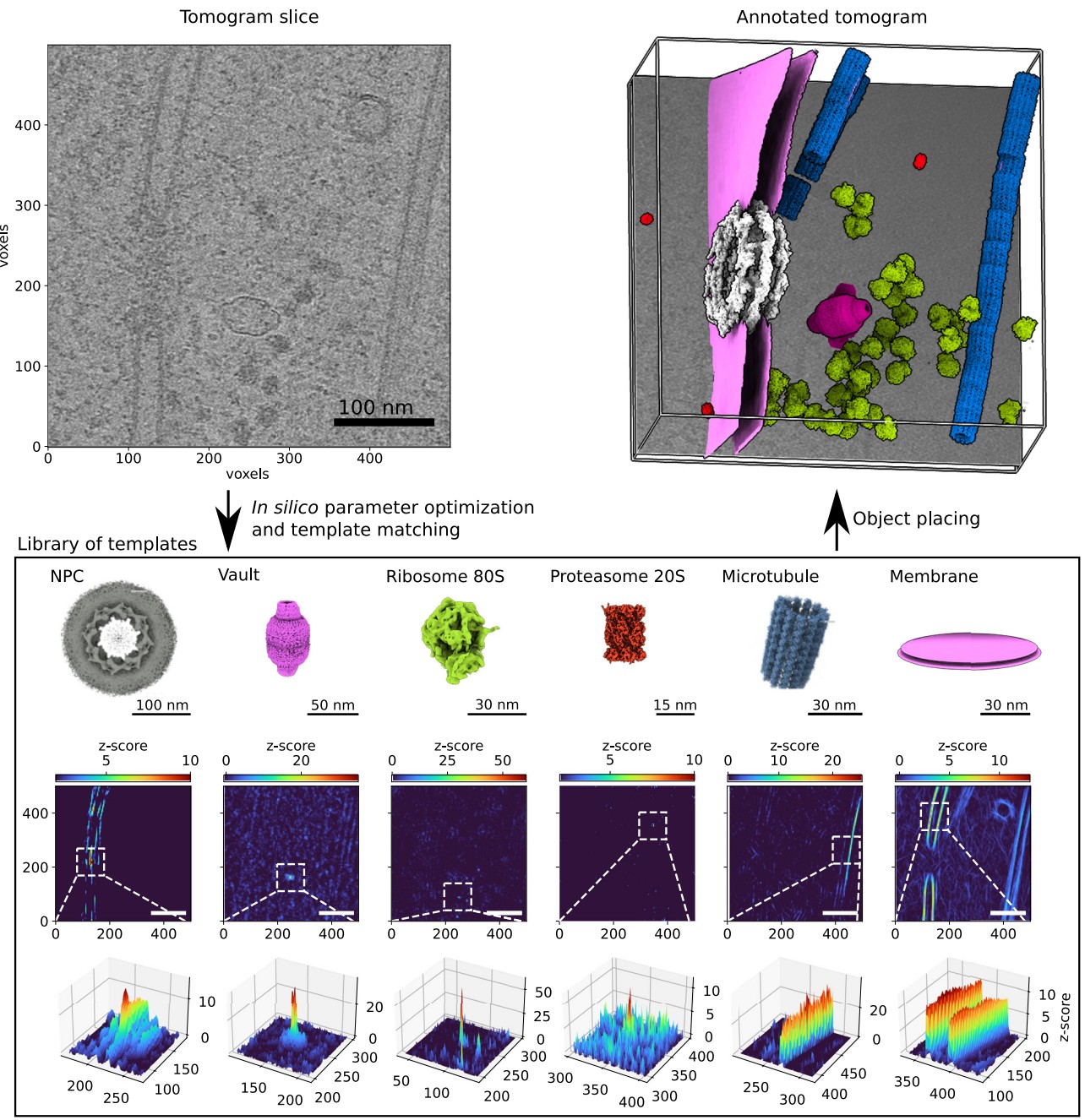

**Fig. 1 | Template matching for visual proteomics.** A tomogram (slice shown in the top left) is cross-correlated independently with each template in the library (bottom) to identify points with high constrained cross-correlation values (zoom-ins with CC z-scores at the bottom). From the z-score maps, 3D localization maps (top right) are generated for visualization[65] and analysis of the spatial interactions of proteins and their complexes. Source data are provided as a Source Data file.

various parameters and found that optimal TM requires systematic tuning of the bandpass filters (Fig. 2a, b), template (Supplementary Figs. 1–4) and mask size (Fig. 2c), voxel size (Supplementary Fig. 5) and angular sampling (Supplementary Figs. 5 and 6). Optimal parameter values depend on the quality of the data as well as the size and shape of the object (Fig. 2d–f and Supplementary Figs. 1–4).

We first assessed the impact of frequency contents. For the ribosome, NPC subunit (C8-symmetric rotational segment), half vault, and microtubule templates, peaks decay with increasing high-pass filter, i.e., when low-resolution information is gradually removed (Fig. 2a). The low-pass filter has a less pronounced effect, although the z-score slightly increased when high-resolution information was included (Fig. 2b). This analysis implies that for ribosome, NPC

subunit, vault and microtubule, TM detection benefits from retaining higher resolution information in the data.

Regarding mask sizes, we found that mask tightness has a negligible effect for ribosomes and microtubules as long as the template is completely contained (Fig. 2c). However, for membrane-associated structures such as the NPC, a shaped mask may exclude the membrane from the template, improving TM performance (Supplementary Fig. 7).

## Angular scanning should be optimized in a template-specific manner

The above analysis indicated that the impact of parameters such as voxel size or the number of orientations sampled depends on the

**Table 1 | Tested cases for template matching**

| Tomograms species | Tested template | | TM parameters | |
|---|---|---|---|---|
| | Name (species) | Reference | Angular step (deg) | pixel size (Å) |
| *D. discoideum* | | | | |
| (Tomograms analyzed: 30) | NPC (*D. discoideum*) | EMD-19139 | 10 | 8.7 |
| Software: STOPGAP and GAPSTOP™ | NPC subunit: C8-symmetric rotational segment (*D. discoideum*) | EMD-19139 | 30, 20, 10, 5 | 8.7 |
| | NPC subunit: C8-symmetric rotational segment (*H. sapiens*) | PDB-id: 7R5J | 10 | 8.7 |
| | Modeled vault (*D. discoideum*) | Swiss model | 10 | 8.7 |
| | Modeled half vault (*D. discoideum*) | Swiss model | 10 | 8.7 |
| | STA vault (*D. discoideum*) | This work[56] | 30, 20, 10, 5 | 8.7 |
| | Ribosome 80S (*D. discoideum*) | EMD-15810 | 30, 20, 10, 5 | 8.7 and 4.35 |
| | Small ribosomal subunit 40S (*D. discoideum*) | This work[56] | 10 | 4.35 |
| | Large ribosomal subunit 60S (*D. discoideum*) | This work[56] | 10, 5 | 4.35 |
| | Unrotated ribosome (*D. discoideum*) | EMD-15812 | 10 | 4.35 |
| | Rotated ribosome (*D. discoideum*) | EMD-15815 | 10 | 4.35 |
| | Microtubule (*H. sapiens*) | EMD-6351 | 10 | 8.7 |
| | Protofilament (*H. sapiens*) | EMD-6351 | 10, 5 | 2.446 |
| | αβ-tubulin dimer (*H. sapiens*) | EMD-6351 | 5 | 2.446 |
| | Proteasome 20S (*H. sapiens*) | EMD-4877 | 10 | 4.35 |
| | Membrane - small STA | This work[56] | 20, 10, 2 | 8.7 |
| | Membrane - large STA | This work[56] | 20, 10, 2 | 8.7 |
| | Membrane from MD - atomistic model | This work[56] | 20, 10, 2 | 8.7 |
| *S. pombe* | NPC subunit: C8-symmetric rotational segment (*S. pombe*) | EMD-11373 | 5 | 13.48 |
| (Tomograms analyzed: 1) | Ribosome 80S (*S. pombe*) | EMD-14426 | 5 | 13.48 |
| Software: STOPGAP | Fatty acid synthase - FAS (*S. pombe*) | EMD-14412 | 5 | 13.48 |
| | Membrane - large STA | This work[56] | 2 | 13.48 |
| *Hek293 cells* | Protofilament (*H. sapiens*) | EMD-6351 | 5 | 4.884 |
| (Tomograms analyzed: 1) | | | | |
| Software: GAPSTOP™ | | | | |

template mass, shape, and size. To systematically investigate this, we developed a Python-based tool to evaluate TM parameters in silico (see details in Methods and examples in Supplementary Figs. 1–4). The in silico evaluation of multiple templates showed that the CC depends almost linearly across different templates on the fraction of overlapping voxels between the rotated template and the object (Fig. 2d, e), a relation that would be exact if voxel intensities were strictly zero or one. The number of overlapping voxels depends on both angular sampling and object shape (Fig. 2e). This effect is particularly pronounced for hollow objects such as the vault and elongated structures such as protofilaments. In such cases, even small rotations lead to a large decrease in the number of overlapping voxels and hence in the cross-correlation. Thus, templates that require finer orientation sampling to be localized with high confidence will demand more computational power for detection with similar performance (Supplementary Figs. 8 and 9). We conclude that general recommendations for sampling during template matching cannot be made. Instead, optimal angular sampling is highly dependent on template shape and should be individually tested. Therefore, our pipeline allows us to optimize parameters in silico in a template-specific manner, prior to analyzing experimental data, to then channel the available computational power toward those templates that require more fine-grained scanning. For example, the variation of cross-correlation with angular distance (Supplementary Figs. 2b, 3b, and 4b) provides an initial guide for estimating axis-dependent angular steps. Angular steps that result in <40% decrease in cross-correlation are considered sufficient, as illustrated for the Vault (Supplementary Figs. 2 and 9) and the NPC subunit (Supplementary

Figs. 4 and 8). Our Python-based tool will allow users to do this systematically for any template.

**Quantitative localization of ribosomes**

Although the qualitative detection of ribosomes was reported[4], reliable particle detection with minimal false negative rates is a prerequisite for quantitative analysis of the localization and interaction of molecular complexes. We assessed the ability of optimized TM to locate individual ribosome positions and orientations by comparing the results of TM with existing annotations of the cytosolic 80S ribosomes for *D. discoideum*[4]. The annotations were obtained in a multi-step classification procedure, with an initially oversampled set of ribosomes, using Relion[40], as described in ref. 4, which resulted in a map with resolutions up to 4.5 Å.

Figure 3 shows the results for TM on 4-binned data (8.704 Å/voxel). Motivated by our in silico evaluation (Supplementary Fig. 5), we assessed the effect of the number of orientations by sampling the rotational space in angular steps of 30, 20, 10, and 5 degrees (576, 1944, 15192, and 119952 orientations) and selected TM peaks corresponding to local maxima in the z-score map that are above a threshold (Fig. 3) and clearly inside the lamella borders. We considered a particle in the ground truth as TM detected if it was located within 10 nm (~1/3 of the ribosome diameter) of a TM peak. With increased numbers of orientations, the z-scores of the peaks increased and with that the percentage of TM-detected particles (Fig. 3c, d; see also Supplementary Figs. 5 and 6). With orientations separated by ~5 degrees, TM detected ~95% of the 437 previously annotated particles with a mean distance to the TM peak of $(3.73 \pm 1.57)$ nm (Fig. 3f) and

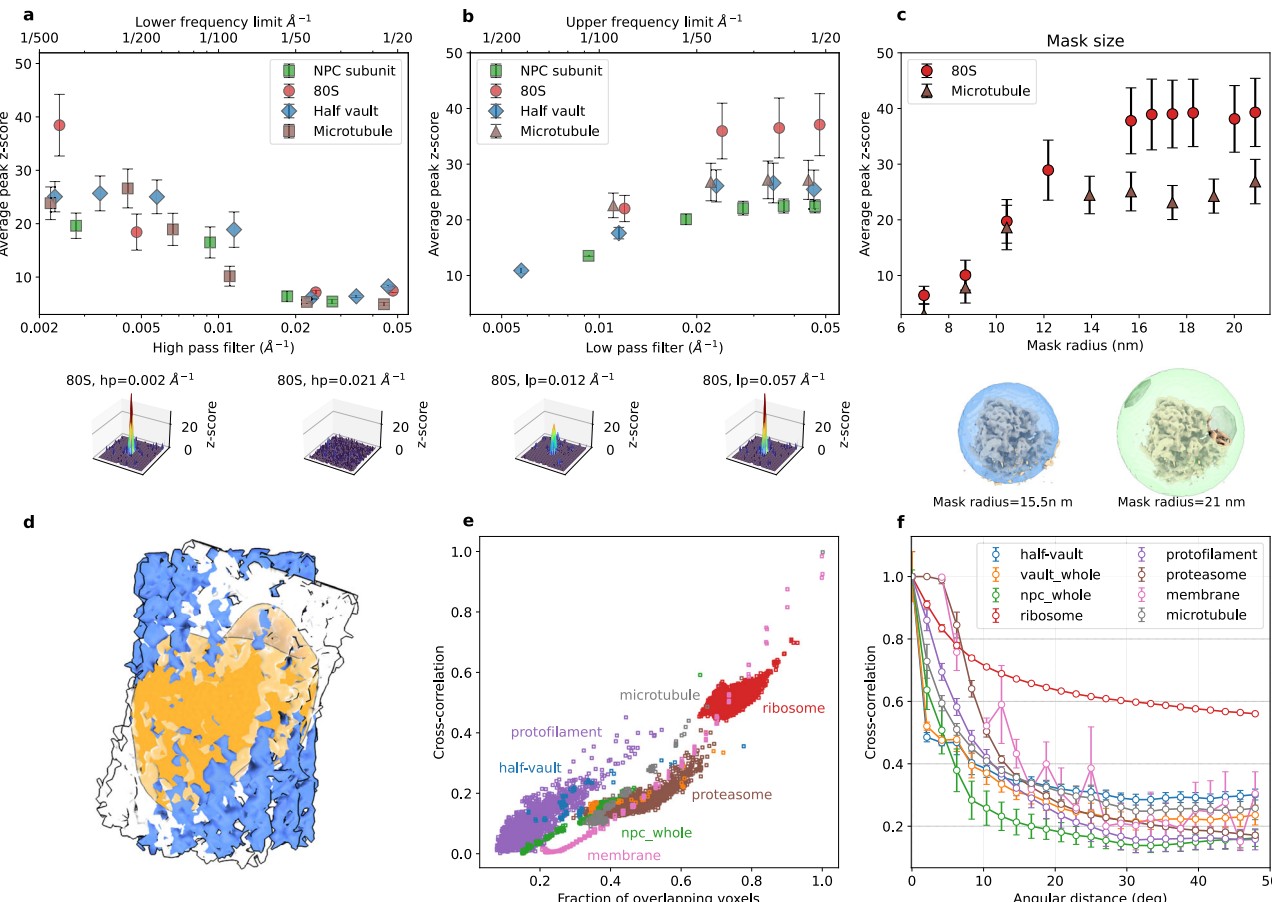

**Fig. 2 | Optimization of the search parameters in template matching.**
**a**–**c** Dependence of the average constrained cross-correlation peak height (z-scores) for 80S ribosomes, NPC subunit, half vault, and microtubule on the high-pass (**a**) and low-pass filters (**b**), and on the diameter of the spherical mask for the ribosome and a cylindrical mask for the microtubule (**c**). In **a** no low-pass filter was applied, and in **b** no high-pass filter. **d** Schematic representation of the overlapping voxels (orange) when a microtubule is rotated around its optimal orientation by 20 deg. **e, f** Dependence of the constrained cross-correlation of a template with itself (in silico evaluation) as a function of the fraction of overlapping voxels and the angular distance (**f**). **e, f** present cross-correlation coefficients (and not z-scores) for the template matched against itself, not the tomogram. In **e** the CC values for all the rotations in a 10 degrees grid are shown, in **f** the average of the CC values for all rotations sampled in a 1-degree grid and binned every 2 degrees. For the average, the number n increases as a function of the angular distance (from $n = 4$ at the pole to up to $n = 306$ at 50 deg). In all cases, error bars correspond to one standard deviation. In **a**–**c** $n = 10, 45, 4$, and 16 for NPC subunit, ribosome 80S, half vault, and microtubule, respectively. Source data are provided as a Source Data file.

with orientations that closely matched the annotated orientations (Fig. 3e). Consequently, the averages of the particles detected and orientated by TM recapitulate the density of the 80S ribosome with high sensitivity and accuracy without the need for a multistep classification process (Fig. 3h, i), similar to recent reports[41]. This suggests that TM can be used for a quantitative accounting of the particles present in the tomograms, whereby false negative detections appear to be minimal. Our analysis shows that the comprehensive search of the rotational space enhances the quantitative capability of TM[41] in a trade-off with increased computational cost.

### High-confidence TM reveals membrane compartments

Accurate segmentation of membranes is crucial for visualizing cellular landscapes, and to the best of our knowledge, TM has not yet been used to detect cellular membranes. We tested TM for membrane segmentation with models of different origins and sizes (Table 1, Figs. 1 and 4). The first template was the map created from a frame in the trajectory of an atomistic simulation of a membrane in explicit water (atomic model). The second and third models were averages of the nuclear envelope obtained by subtomogram averaging with diameters of 43.5 nm (small STA) and 87 nm (large STA), respectively. For comparison, during TM, cylindrical masks with a diameter of 34.8 nm

were used for both the atomistic and the small STA, while a cylindrical mask with a diameter of 76.5 nm was used for the large STA (see "Methods" section).

The inner and outer membranes of the nuclear envelope were detected using any of the three templates (atomistic, small STA, large STA; see Supplementary Movie 1). The atomistic and small STA templates performed roughly on par. Increasing the number of orientations (20, 10, and 2 degrees at 4-binned data with 8.704 Å/voxel) consistently decreased the background noise (Fig. 4), sharpening the peaks, and increased the confidence in the TM detection. False positives for the small templates (atomistic, small STA), e.g., from a microtubule segment (Fig. 4 left; see also Fig. 1) are suppressed by using the large STA template (or, visually, by recognizing the lacking 2D extension). However, the large STA model gives only a weak signal for curved membranes, pointing to the need for an expanded model set of membrane patches of varying curvature.

Although computationally expensive compared to other segmentation methods, template matching for membranes has several strengths. For example, the template matching output could be used as an initial annotation for training deep-learning algorithms. In addition, TM not only predicts the positions of the membranes in the

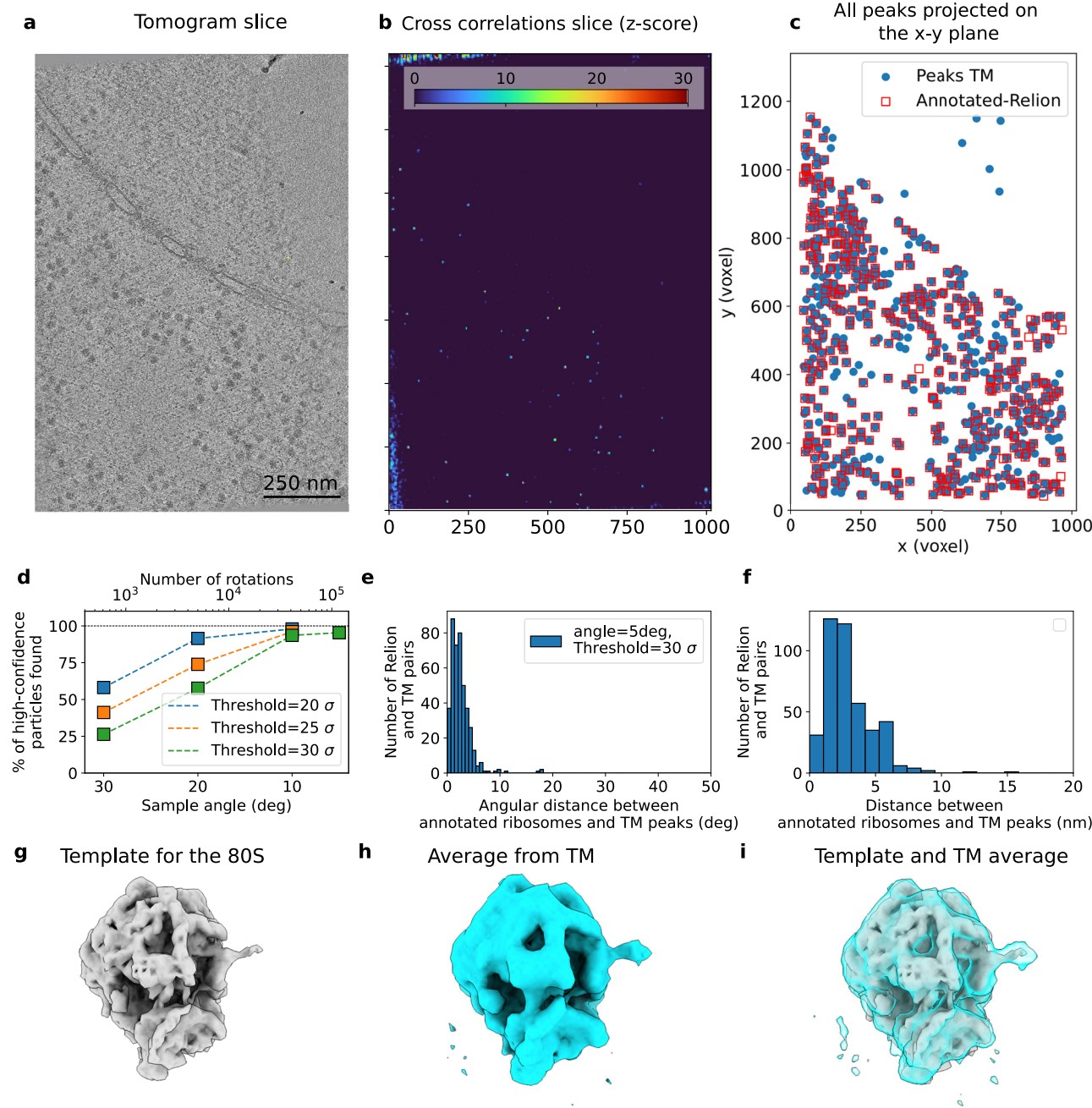

**Fig. 3 | Template matching locates the 80S ribosome with high spatial and rotational accuracy. a** Tomogram slice showing abundant ribosomes. **b** Slice of the z-score map obtained from template matching using the template of the 80S ribosome shown in (**g**). **c** Superimposition of the peaks obtained from template matching (blue circles; sampled every 5 degrees and with a cross-correlation threshold z-score ≥ 30) and the high-confidence localizations obtained from an expert multiple-step alignment using Relion[40] reported in ref. 4. **d** Percentage of the

high-confidence Relion particles detected within 10 nm of the TM peaks as a function of the rotational sampling (i.e., number of orientations). **e, f** Histograms of angular (**e**) and Euclidean distances (**f**) from the TM peaks to the annotated Relion particles, respectively, each obtained for a 5-degree angular sampling. **g** Template for 80 S ribosome. **h, i** Ribosome structure obtained by averaging the particles from TM from (**c**) (**h**, no further processing) superimposed on template (**i**). Source data are provided as a Source Data file.

tomogram but also provides voxel-by-voxel normal vectors, which in turn enables a detailed analysis of the local properties of the membranes. The latter could also be used as an automatic input for triangulation methods and/or as a starting point for simulations of membrane dynamics.

**Detection of subunits and conformer subpopulations**

We tested the ability of TM to localize subunits and assign substates of ribosomes, the NPC, and microtubule fragments. We generated templates for the subunits of the *D. discoideum* NPC according to its C8

symmetry, microtubule protofilaments, the small (40S) and large (60S) ribosomal subunits, and for two prominent 80S ribosome states capturing the ratchet-like motion essential for protein synthesis[42].

For the ribosomal subunits, we performed TM on 2-binned data (4.352 Å/voxel) with orientations every 10 degrees, since TM on 4-binned tomograms showed inconclusive peaks. A sub-volume of the tomogram was analyzed independently with three different templates: 80S, 60S, and 40S (Fig. 5). Similar to the 4-binned data (Fig. 3), the TM localized 96.9% of the 80S annotated ribosomes with z-score peaks up to 114 (Fig. 5b, c). Furthermore, when comparing the positions and

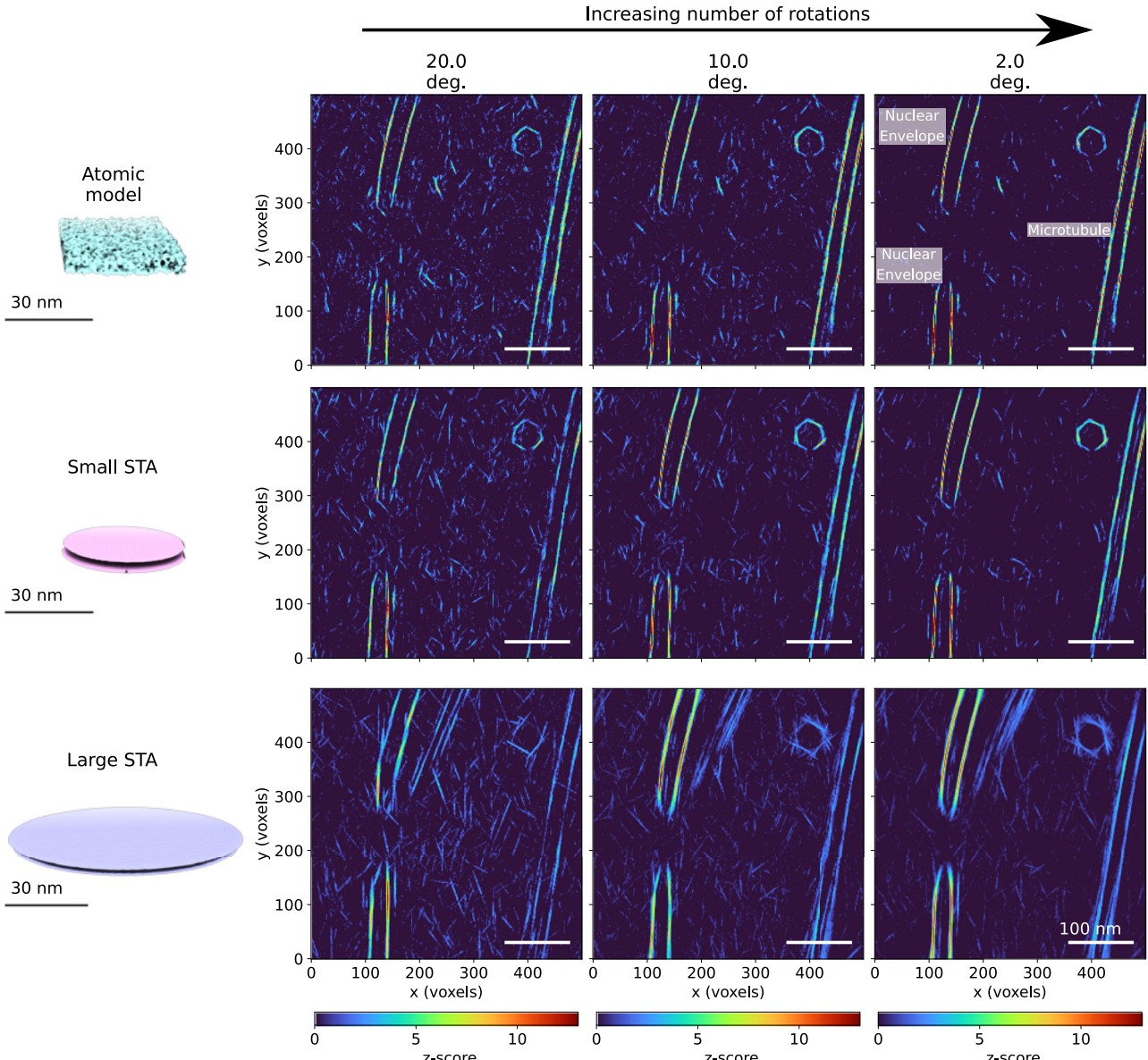

**Fig. 4 | Template matching for the segmentation of membranes in 3D.** Results in the top, middle, and bottom row were obtained for templates constructed from a simulated atomic membrane and STAs of the nuclear membrane with diameters of 50 (small STA) and 100 (large STA) voxels, respectively (8.704 Å/voxel). The results from left to right correspond to increasing angular sampling of 20, 10, and 2 deg, respectively. Note that the peaks at the upper right corner originate from a highly curved vesicle while the two stripes on the right-hand side of the cross-correlation maps (z-scores) are a microtubule and not a membrane (see Fig. 1). Source data are provided as a Source Data file.

orientations of the subunits, TM correctly predicted the location of the subunits and their relative orientations (Fig. 5d). Small but noticeable differences between the orientations of the subunits with respect to the position of the 80S reflect the limited angular sampling. Finally, using all the TM peaks detected by the 60S (90 particles) or only the unannotated (34 particles) TM peaks detected by the 60S, we recovered features from the 80S (see Supplementary Fig. 10), demonstrating the high quality of the particles found.

By comparing the relative TM z-scores on 2-binned data (4.352 Å/voxel) with orientations every 10 degrees, we could correctly assign the ratcheting state of the small subunit of individual ribosomes in space (Fig. 5e–h). Two known representative ratcheting states of the *D. discoideum* ribosome were used as templates[4]: rotated (EMD-15815 https://www.emdataresource.org/EMD-15815) and unrotated (EMD-15812 https://www.emdataresource.org/EMD-15812), and the states were assigned using the expectation-maximization algorithm (see

"Methods" section for details) to predict the mixture of subpopulations (Fig. 5g), similar to previous studies[43]. Although the rotated and unrotated templates share most of the density with only a slight rotation of the 40S (Fig. 5f), the TM assignments differentiated between the rotated and unrotated states, matching the existing annotations in 77.7% and 82.4% of cases, respectively (Fig. 5e, h). It is worth noting that there are other intermediate rotation states, and the binding of multiple cofactors to the ribosome along the translation cycle[4] may affect the TM z-scores and ultimately the state assignment, which may account for non-matching particles.

TM also finds NPC subunits. Directly from the z-score maps, we could detect the C8-symmetric rotational segments of the NPC (Fig. 6a) with high confidence, as demonstrated by performance metrics (Supplementary Fig. 7), after performing TM on 4-binned tomograms (8.704 Å/voxel) and sampling orientations every 10 degrees as suggested by our in silico analysis. Interestingly, no peaks

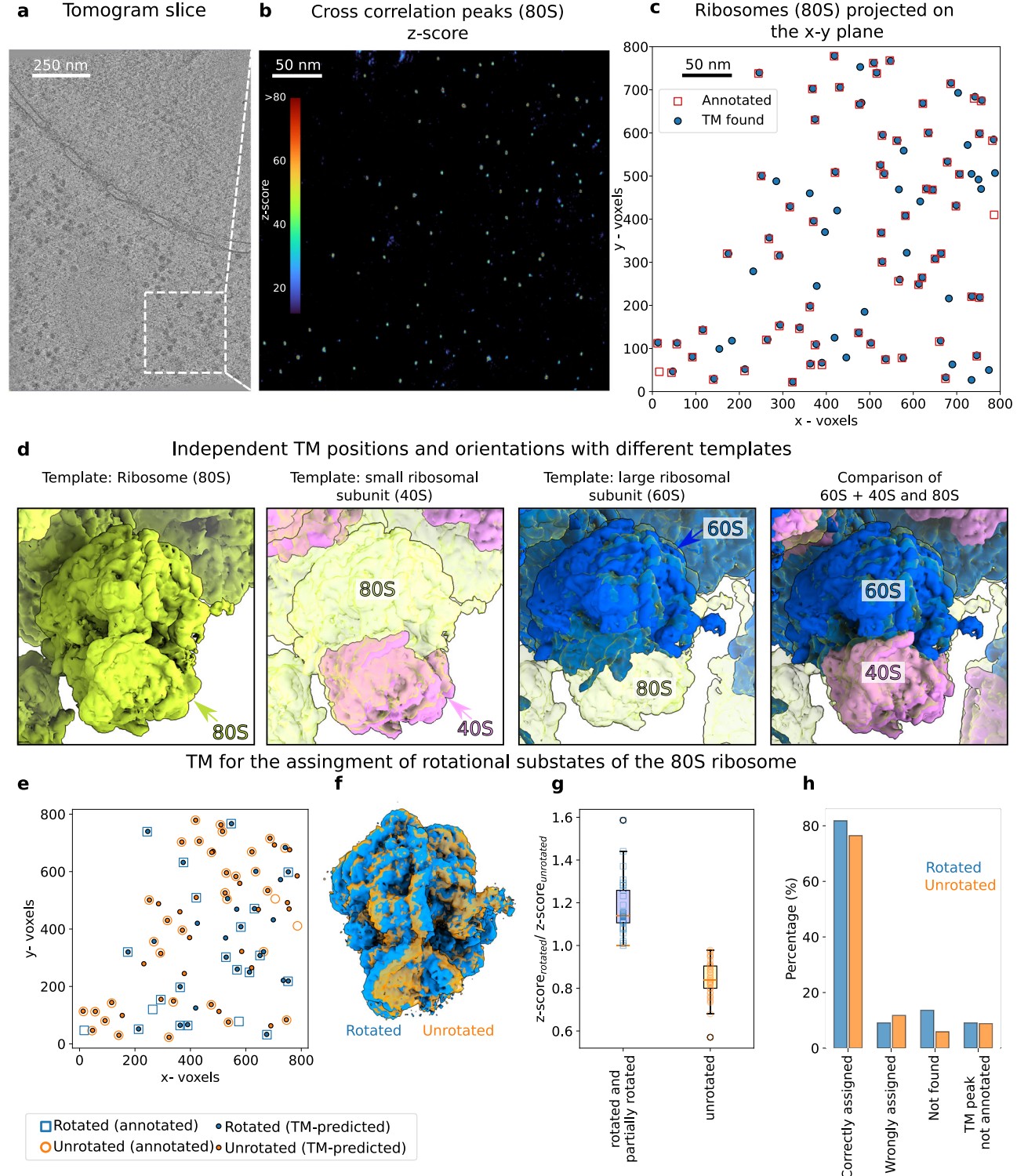

**Fig. 5 | Template matching predicts the relative orientations of ribosome subunits and assigns ribosome rotational States. a** Tomogram slice. **b**, **c** TM finds the positions of the 80S (green) ribosomes annotated with high confidence (96.9% of particles within 2.4 ± 1.5 nm). Two more templates were tested, (i) the small (40S - pink) and (ii) the large (60S - blue) ribosomal subunits. **d** Calculated position and rotation of the 40S and 60S mapped back to the tomogram compared to the calculated orientation for 80S from TM. Note that the positions and orientations displayed the subunits and the 80S were obtained from independent TM calculations. **e** Comparison of the assignment of the ratchet-like rotational states of the ribosome from TM using a Gaussian mixture model[43] with existing annotations[4]. **f** Templates for rotational states. **g** Ratio of TM scores as function as function of assigned rotation state (line: median; box: interquartile range; error bars: range, $n = 28, 46$ for rotated and unrotated, respectively). **h** Consistency of assignment (annotated as rotated: blue; unrotated: orange). Source data are provided as a Source Data file.

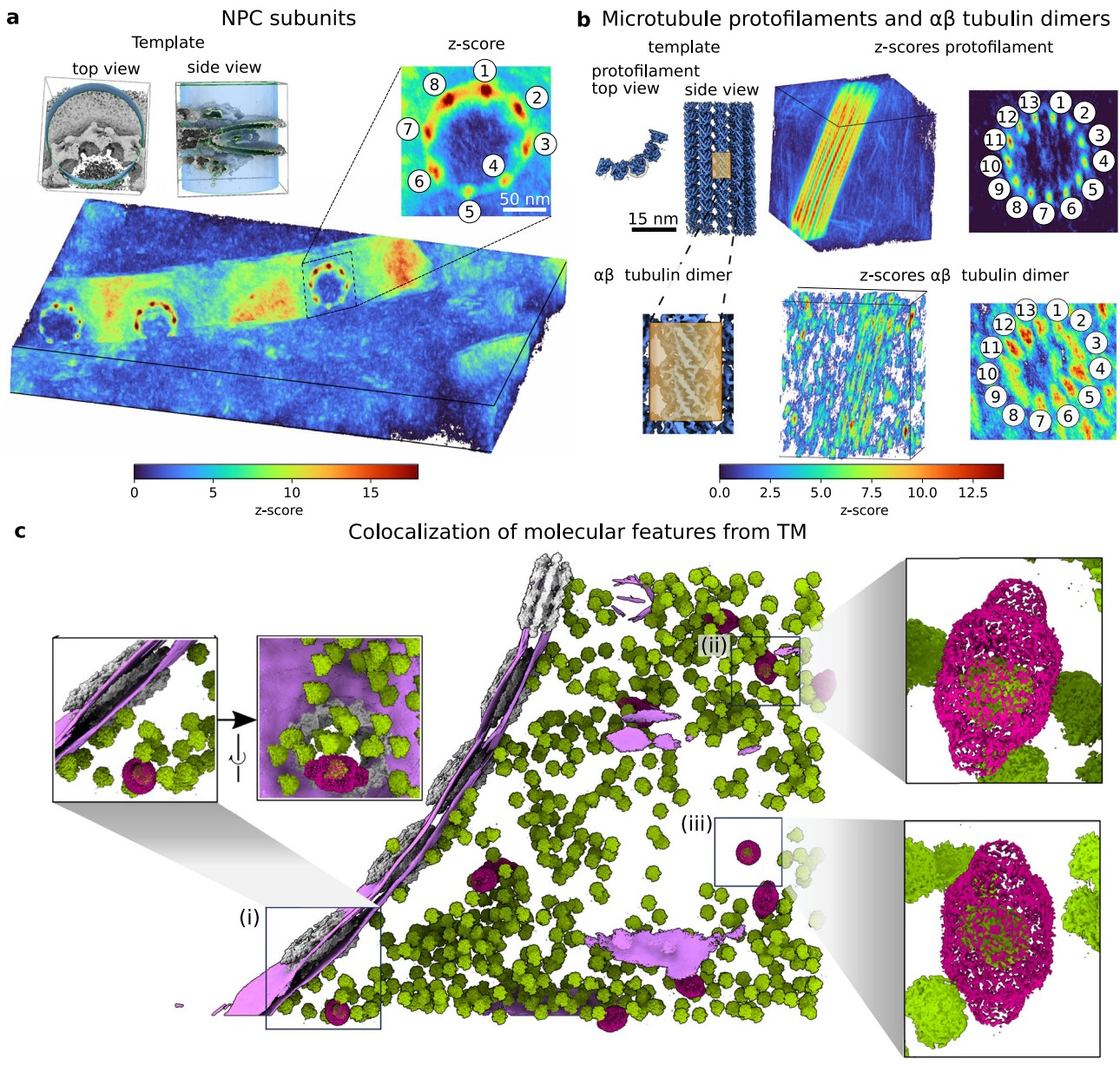

**Fig. 6 | Template matching detects NPC subunits, microtubule protofilaments and ribosome-loaded vaults. a**, **b** Perspective view of the 3D constrained cross-correlation map obtained from template matching using an NPC subunit (**a**) and microtubule section (**b**). The templates are shown on the left-hand side of each panel. For the microtubule, two templates were used: a protofilament and an αβ-tubulin dimer. The 8- and 13-fold symmetries for the NPC and the microtubule, respectively, emerge naturally from template matching (zoom-ins with numbered peaks). Note that two NPCs at the edge of the lamella have only 7 (left) and 5 (center) detectable subunits left after the milling process. The NPC subunit and protofilament templates were cut from the whole NPC and microtubule templates, respectively. The αβ-tubulin dimers in (**b**) were obtained by masking the proto-filament template (orange box). **c** From the 3D localization maps generated for visualization[4] and analysis, TM finds ribosomes inside vaults indicated by the squares (i, ii and iii clockwise from left), see also Supplementary Fig. 14. In (i), the vault (magenta; z-score=54) containing the ribosome (green; z-score=63) is near a NPC (gray) and the nuclear envelope (purple). In (ii) and (iii), the vaults (z-score=32 and z-score=59) containing the ribosomes (z-score=46 and z-score=77) are in the cytoplasm. The templates used for the ribosome and the vault are identical to those used in Fig. 1. Here the vault is shown with a lower threshold level to visualize the ribosome on the inside.

were detected using an NPC from a different species as a template (Supplementary Fig. 7), highlighting the role of template information content.

To investigate the effect of template size, we used segments of microtubules differing in size. Using appropriate sampling (2-binned, 2.446 Å/voxel, 5 degrees), TM resolved peaks of the individual αβ-tubulin as distinct peaks with the 13-fold symmetry of microtubules (Fig. 6b) when the protofilament template was used. This is apparent in tomograms of both, *D. discoideum* and Hek293 cells (Supplementary Fig. 11). We further masked a single αβ-tubulin dimer (Fig. 6b and Supplementary Fig. 12). Despite the low combined mass of only 100 kDa, TM achieves good statistics both in terms of true positives and (likely) false negatives. Although the subunit segmentation along the filament was blurred, it is evident in the longitudinal z-scores along the axial lines passing through protofilaments that the local maxima correspond to the subunits in the microtubule lattice for both the protofilament template and the αβ-tubulin dimer (Supplementary Fig. 12). When cylindrical segments of different sizes are used as template, microtubules are detectable at lower resolution (8.704 Å/voxel, 10 degrees), but the

true positive rate is reduced with decreasing template size (Supplementary Fig. 13).

Overall, these results demonstrate that TM can find subunits of macromolecular complexes with high accuracy and precision.

## High-confidence TM identifies vault-encapsulated ribosomes in situ

The biological function of the vault particle remains mysterious. A few interactors binding to the inside surface have been reported[44,45] which in line with its capsule-like morphology has led to speculations that vaults may enclose other particles and transport cargo within the cell. To the best of our knowledge, however, evidence for vaults encapsulating cargo in situ is yet missing. Three of the vaults in the tomogram of Figs. 5a and 6c contain 80S ribosomes with highly significant z-scores (vaults: 54,32, and 59 ribosomes: 63, 46, and 77, in Fig. 6c(i)–(iii), respectively). Note, that TM reports excellent performance metrics for the identification of ribosomes and vaults (Supplementary Fig. 14). These findings support the hypothesis that vaults can be cargo-loaded in situ. Whether the encapsulation occurred during vault biogenesis or by transient opening remains to be further investigated.

## High-confidence TM identify macromolecular complexes in other species: comparison with state-of-the-art tools

We further tested the versatility and performance of high-confidence TM on a recently published tomographic dataset of *S. pombe*[30]. We selected this dataset because it was used to test two recent deep-learning tools to localize particles in tomograms (DeePiCt[30] and DeepFinder[29]) and annotations exist. We performed TM for ribosomes (80S), fatty acid synthase (FAS), membrane, and NPCs on a tomogram reconstructed from the tilting series reported for *S. pombe* (EMPIAR-10988 [https://www.ebi.ac.uk/empiar/EMPIAR-10989/];TS_043)[30] (see Methods and Supplementary Fig. 15). Templates for *S. pombe* ribosomes (EMD-14426 [https://www.emdataresource.org/EMD-14426])[30], FAS (EMD-14412 [https://www.emdataresource.org/EMD-14412])[30] and the NPC (EMD-11373 [https://www.emdataresource.org/EMD-11373])[46] were obtained from the EMDB[34]. For the membrane we used the large STA template described above (see Fig. 4). From the whole NPC template, a smaller template of a rotational segment was extracted in a procedure analogous to Fig. 6a. TM was performed on one 4-binned tomogram (13.48 Å/voxel) for all the templates. We used angular steps of 5 degrees for the 80S, FAS, and NPC subunits, and 2 degrees for membranes.

For the ribosome localization, TM had an F1 score of 0.77, which is comparable to DeepFinder (median F1 = 0.83) and DeePiCt (median F1 = 0.79). TM performs significantly better on FAS (F1 = 0.70, Supplementary Fig. 15) than DeepFinder (median F1 = 0.11) and DeePiCt (median F1 = 0.46). Finally, in contrast to DeepFinder and DeePiCt that faced challenges locating the NPC, TM demonstrated its capability to precisely identify the individual NPC subunits with z-scores >20 (Supplementary Fig. 15d), as confirmed by expert inspection. TM also localized membranes with a generic, unadjusted template. All results for DeepFinder and DeePiCt were taken from ref. 30.

## Discussion

The comprehensive identification of particles in electron tomograms remains challenging. Despite its conceptual simplicity, template matching has been considered a low-precision method, and its application has been limited by the low signal-to-noise ratio of tomographic data, the scarce availability of suitable templates, and the lack of objective optimization of search parameters. Here, we have shown that template matching can identify the positions and orientations of multiple macromolecular complexes in living cells with high accuracy and fidelity. For this task, templates can be used from multiple sources such as data banks, simulations, homology modeling, or volumetric data from the tomograms. For maximum efficiency, we developed

software for an in silico parameter optimization and GPU-accelerated Python Stopgap for template marching (GAPSTOP™).

With optimized TM, we achieved a mass resolution of 100 kDa in experimental tomograms of a crowded cell. Using a generic template for human tubulin, we could readily localize individual tubulin subunits in a high-resolution CryoET map of *D. discoideum* cells (Fig. 6b, Supplementary Fig. 12). High-confidence TM thus pushes into a particle size regime in situ that covers much of cellular biology.

By exploiting geometric and contextual features, one can further improve the likelihood of finding objects by template matching. Vaults, for example, are of low abundance with a low-contrast interior, but their unique shape facilitates identification with confidence (Fig. 1 and Supplementary Fig. 2). Spatial extent is also important for TM. Another strategy to search for smaller objects is to decrease the voxel size (the same object will occupy more voxels, increasing the range of frequencies describing the template), which allowed us to locate ribosomal subunits and distinguish between ribosomal substates. However, the location of smaller isolated objects poses an additional challenge: the unambiguous validation of the peaks. In the cases presented here, we used annotated data (Figs. 3, 5 and Supplementary Figs. 5, 14, and 15) and expert inspection (Figs. 1, 4, 6 and Supplementary Figs. 7–9 and 11–16). However, when considering smaller structures, an increase in the number of peaks and volumes of high significance is expected. To overcome this challenge, we envision a hierarchical approach in which we mask parts of the volume where we have high confidence in the presence of an object and then perform a focused search for smaller objects. Still, relying on template matching alone may be insufficient, and additional information, such as abundance data would need to be incorporated to effectively analyze TM results.

High-confidence TM outperforms existing deep-learning-based tools in the reliable detection of challenging low-abundance and low-density complexes (e.g., NPC). TM delivers competitive or superior statistics compared to recent approaches based on artificial neural network architectures[29,30]. Unlike existing deep-learning-based approaches[29–31], TM does not require any prior training, which makes it possible to precisely localize subunits and identify functional substates by distinguishing between multiple conformations (Figs. 5 and 6 and Supplementary Fig. 12). However, the widespread use of template matching was limited by its computational expense, due to the nature of the algorithm that evaluates each voxel in the volume and the need of extensive angular sampling (up to hundreds of thousands of orientations)[41]. This problem was exacerbated in the current STOPGAP implementation with limited parallelization across CPUs. However, the TM algorithm is by construction embarrassingly parallel and relies on Fourier transformation, which is highly efficient on graphics processing units (GPUs). Therefore, we developed and released GAPSTOP™ to harness the GPU's parallel capabilities for TM while preserving the unique missing wedge and noise correlation modeling of the STOPGAP implementation[36]. Significant speedups by using GPUs were also reported for pytom_tm[41].

Finally, the TM workflow can readily be combined with AI-based approaches[22,28–31]. At one end of the pipeline, AI can be used to optimize TM parameters and, at the other end, to integrate the outputs across template families into classification scores. At the center of the pipeline, however, the 3D CC score is highly efficient and captures the relevant physics by being rigorously proportional to the log-likelihood for Gaussian noise in the 3D map[47] (see "Methods" section). In the future, TM-annotated tomograms can be used to train and validate AI-based particle localization methods.

Taken together, our analysis demonstrates the detection of various objects, with high confidence, in cryo-electron tomograms acquired with the latest hardware. By expanding the repertoire of templates, e.g., from AlphaFold[48] and molecular dynamics simulations, TM should help us assign molecular identities to the large parts of

tomograms currently unassigned. High-confidence TM thus changes the workflow in CryoET through fast, automated, objective, and comprehensive feature identification. In turn, CryoET combined with high-confidence TM brings us closer to the goal of visual proteomics: to map the positions and orientations of all macromolecular complexes within living cells.

## Methods

### Experimental tomograms

The tilt series of *D. discoideum* used in this study, as well as the annotations for the ribosomes and their substates were previously reported (codes: EMPIAR-11845 and EMPIAR-11899)[4]. The tilting series for *S. pombe* were obtained from the Electron Microscopy Public Image Archive (EMPIAR)[49], code: EMPIAR-10988[30]. The tilting series for *H. sapiens* have EMPIAR code: EMPIAR-11538. For the three species, the cell culture, sample preparation, data acquisition, and image processing are detailed in the original publications. For *D. discoideum*, in brief, tilt series were collected at 300 kV on a Titan Krios G2 microscope equipped with a Gatan BioQuantum-K3 imaging filter in counting mode and a Titan Krios G4 microscope equipped with a cold FEG, Selectris X imaging filter, and Falcon 4 direct electron detector in counting mode. Projections had a pixel size of 2.176 Å and 1.223 Å for *D. discoideum*, 3.37 Å for *S. pombe*, and 1.223 Å for *H. sapiens* respectively, and were acquired in a dose symmetric acquisition scheme[50] with 2 deg increments[4,51]. For *D. discoideum*, *S. pombe,* and *H. sapiens*, the initial tomogram reconstruction was performed in eTomo from IMOD[52], and the established parameters were used to reconstruct the tomograms with 3D-CTF correction using novaCTF[53]. The corrected tomograms were used for TM either in their unbinned form or with applied binning of 2, 4, or 8.

### Template matching

We performed TM using STOPGAP[35] and GAPSTOP™ (GPU-Accelerated Python STOPgap for Template Matching) for the cases described in Table 1. STOPGAP is an open-source freely available Matlab-based code: https://github.com/williamnnwan/STOPGAP. GAPSTOP™ is a Python implementation of the STOPGAP framework that speeds up TM by 10–100 times through GPU utilization. Fully implemented in Python, GAPSTOP™ is now available to the community via its repository: https://gitlab.mpcdf.mpg.de/bturo/gapstop_tm. Documentation and installation instructions are provided for ease of use (https://bturo.pages.mpcdf.de/gapstop_tm).

As input, both STOPGAP and GAPSTOP™ require a template, a list of orientations to probe (angular sampling), a wedge list, definitions of the filters, and the reconstructed tomogram. Details on the preparation of the templates are given below. The list of orientations was generated using STOPGAP function generate_angle_list, which samples the angle space uniformly on a grid and also takes into account the symmetry of the template (see Table 1). The wedge list contains the acquisition parameter information for the individual tilts. In particular, it must contain: the pixel size, the tilt angle, defocus, and electron dose. The low-pass filter allows low-frequency signals to pass through while attenuating high-frequency signals, while high-pass filters allow high-frequency signals to pass through while attenuating low-frequency signals. In STOPGAP, both are defined in voxels defining the radius of a spherical mask applied in Fourier space (i.e., these values depend on the dimensions of the template). See further details in the STOPGAP documentation. For each template, we obtained a map of the local CC maxima over orientations, which we turned into z-scores as $z = (CC - \mu)/\sigma$ with $\mu$ and $\sigma$ the average and standard deviation of CC values across the map, respectively.

### In silico peak analysis

The template weighting and CC calculation methods from STOPGAP[35] have been ported to Python and extended to output additional information relevant to the input parameters. The inputs are the same as for the original TM, but instead of a whole tomogram, a small volume is used. The volume can be either the same as the template (typically an STA map or a model) or a subtomogram (obtained either based on an existing ground truth or by manual picking). For full peak analysis, one must also provide a density mask, which is a binary (or tapered) map corresponding to the density of the template (or alternatively a threshold to create one during the analysis). In addition to the z-score map and the angles map, the peak analysis provides information on the TM progress as well as the analysis of the template and the resulting maps. A table shows the dependence of the template orientation on the CC scores and on the number of overlapping voxels. For the template, it computes the dimensions, the number of voxels in the density mask, and a solidity calculated as the number of voxels in the density map divided by the volume of its convex hull. It also returns the value of the peak, its exact location, and line profiles through the peak along each dimension. The angle map is used to compute three maps of angular distances, where each voxel contains the angular distance in degrees between the orientation encoded in the angle map and the starting template orientation. The first map contains the angular distance of the full orientation and is computed using a quaternion-based cosine similarity formula[54]. The second map contains the angle between the normal vectors of the final and the initial orientation, which encodes the rotation on the cone. The third map contains the angle between the in-plane vectors. The maps provide information on whether the CC scores are more sensitive to cone or in-plane rotation (or neither) and thus can be used to determine sufficient angular sampling. Finally, the key results of the peak analysis are summarized in a PDF file to provide an easy-to-read overview for the user. While the tool is most useful for determining the optimal setup for GAPSTOP™ (or deciding its feasibility), it can also be used to analyze the origin of false positive results by testing a template against a map containing a different structure. For example, a ribosome template can be tested against map containing proteasome to determine the pixel size and filtering to distinguish these two with sufficient confidence. Similarly, the membrane template can be used against a microtubule structure to determine the size of the template and mask necessary to pick mostly membranes. Lastly, there is a possibility to turn off the missing wedge weighting to analyze its impact on the peak shape or add an angular offset to the starting orientation to see how it affects the peak value for given angular sampling.

### Membrane templates

For the atomistic membrane template, we used the final lipid bilayer of a 28-ns molecular dynamics simulation of a $40 \times 40$ nm$^2$ membrane patch (7164 lipids) in explicit water, using the setup and protocol of ref. 55. Input files and the final output structure can be found in the public repository[56]. The small STA and large STA models were obtained as subtomogram averages of the nuclear envelope with diameters of 43.5 nm and 87 nm, respectively. In TM, cylindrical masks with a diameter of 34.8 nm were used for the atomistic and small STA models. For the large STA model, the diameter was increased to 76.5 nm.

### Creation of density maps from atomic models

In the cases where an atomic model was available (membrane and vault), we used the molmap function of ChimeraX[57] with a resolution of 3.5 Å. Here, each atom is represented by a 3D Gaussian. The width of the Gaussian is given by the resolution, while the amplitude is proportional to the atomic number. Afterward, we used EMAN2[58] to rescale and resample the map to match the voxel size of the respective tomogram.

### Templates for the ribosomal subunits 40S and 60S

To generate templates of the large 60S and small 40S ribosomal subunits, the ribosome structure of a translating *D. discoideum*

ribosome from EMD-15810[4] was segmented in ChimeraX[57] using the Segger function[59]. A fitted eukaryotic ribosome atomic model (PDB-id: 5LZS[60]) was used to guide this procedure.

### Statistical assignment of ribosomal substates

To assign the substates of the ribosome 80S, we performed TM using two templates corresponding to a rotated (EMD-15815 [https://www.emdataresource.org/EMD-15815]) and unrotated ribosomal states (EMD-15812 [https://www.emdataresource.org/EMD-15812]). High-confidence peaks were extracted from each TM map with their respective z-scores.

For a given particle, defined by its coordinates, we computed the ratio between the two TM z-scores as:

$$x_i = \frac{cc_{i-rot}}{cc_{i-unrot}} \tag{1}$$

where $cc_{i-rot}, cc_{i-unrot}$ corresponds to the z-score obtained for particle $I$ with the rotated and unrotated template, respectively. To assign the rotational substates of each particle, we used a Gaussian mixture model (GMM). Specifically, we assumed that the distribution of $\mathbf{x}$ can be modeled as a linear superposition of two Gaussian distributions, one for the rotated state and the other for the unrotated state. The probability density function of the GMM can be written as:

$$p(\mathbf{x}) = \pi_{rot} \mathcal{N}(\mathbf{x}|\mu_{rot}, \Sigma_{rot}) + (1 - \pi_{rot})\mathcal{N}(\mathbf{x}|\mu_{unrot}, \Sigma_{unrot}) \tag{2}$$

where $\mathcal{N}(\mu, \Sigma)$ represents a Gaussian probability density function with mean $\mu$ and variance $\Sigma$.

To estimate optimal parameters for $\pi_{rot}, \mu_i, \Sigma_i$, we used the *sklearn.mixture*[61] python implementation of the estimation-maximization (EM) algorithm. The EM algorithm alternates between computing the expected values of the latent variables (the assignment of each data point to a mixture component) and updating the parameters of the GMM to maximize the log-likelihood of the observed data. Specifically, the E-step computes the posterior probability of each mixture component for each data point, given the current estimates of the parameters, while the M-step updates the parameters to maximize the expected complete log-likelihood of the data, given the posterior probabilities.

### Determination of the receiver operating characteristic (ROC) curves and F₁ scores

Adjusting the thresholds for the z-score derived from template matching results in varying balances between specificity and sensitivity. This trade-off can be depicted in a graphical representation known as a receiver operating characteristic (ROC) curve, as detailed in ref. 62.

The ROC curve illustrates sensitivity (true positive rate - TPR) on the y-axis and $1 -$ specificity (false positive rate - FPR) on the x-axis. The area under this curve provides a concise metric summarizing the classifier's overall performance.

$$\text{Sensitivity or true positive rate} = \frac{TP}{FP + TP} \tag{3}$$

$$\text{False positive rate} = \frac{FP}{FP + TN}. \tag{4}$$

Another common measure of predictive performance is the F₁-score.

$$F_1 \text{score} = \frac{2TP}{2TP + FP + FN} \tag{5}$$

To generate ROC curves and obtain F₁ scores for ribosomes, NPC, vault, and microtubules, we compared peaks from the TM with

annotated particles. We considered peaks from the template matching with z-scores > 5 and separated by half the diameter of the template. For the 80S ribosomes, we used the annotated particles from a previous publication obtained with Relion classification[4] as reference. Ground truth for the vault and NPC was obtained by expert manual picking followed by subtomogram averaging. For microtubules, a set of ground truth peaks was obtained by expert manual annotation. A particle was considered found if the TM peak was closer than 10, 10, 15, and 5 nm to the annotated ground truth for ribosomes, vaults, NPC subunits (C8-symmetric rotational segment), and microtubules, respectively.

We varied the thresholds on the z-scores for the calculation of true positive rates (sensitivity) and false positive rates. These rates were plotted against each other to construct the ROC curves and the maximum F₁ score, which illustrate the performance of the template matching method across different structures.

### Template matching produces maximum-likelihood solution

For Gaussian noise of width $\sigma$ in the intensities of a CryoET 3D map $M$, the likelihood $L$ that a feature in the map is consistent with a template $T$ rotated and translated by $R$ is proportional to

$$L \propto \exp\left[-\frac{\left(\sum_{i,j,k} \left(M_{ijk} - (RT)_{ijk}\right)^2\right)}{2\sigma^2}\right] \tag{6}$$

By multiplying out the square, summing over the voxels $i,j,k$, and recognizing that the "$M^2$" and "$(RT)^2$" terms are constant, we find that

$$L \propto \exp\left[\frac{(\sum_{i,j,k} M_{ijk}(RT)_{ijk}}{\sigma^2}\right] \tag{7}$$

The term in the exponent is exactly the cross-correlation CC between map and template divided by $\sigma^2$. Analogous to single-particle 2D images[47], the cross-correlation of template and map is thus the log-likelihood scaled by the squared noise amplitude. CC optimization over template rotation and translation $R$ thus gives the maximum-likelihood solution.

### Reporting summary

Further information on research design is available in the Nature Portfolio Reporting Summary linked to this article.

## Data availability

The previously published structures for the NPC subunits (*H. sapiens*) EMD-14325, EMD-14328 and EMD-14330, the NPC (*S. pombe*) EMD-11373, the 80S ribosome (*D. discoideum*) EMD-15810, EMD-15812, and EMD-15815, the 80S ribosome (*S. pombe*) EMD-14426 [https://www.ebi.ac.uk/emdb/EMD-14426], the fatty acid synthase (*S. pombe*) EMD-14412, the 20S proteasome (*H. sapiens*) EMD-4877 and the microtubule (*H. sapiens*) EMD-6351 are accessible through the Electron Microscopy Data Bank. The previously published tilt series for *S. pombe*, EMPIAR-10989 are available through the Electron Microscopy Public Image Archive. The previously published structures 7R5J (NPC structure), 6RGQ (human 20S proteasome structure), and 3JAR (microtubule structure) are available through the Protein Data Base. Source data are provided with this paper. Molecular dynamics setups, templates generated in this study, and supplementary raw data have been deposited in Zenodo (https://doi.org/10.5281/zenodo.10819130)[56]. Source data are provided with this paper.

## Code availability

All code used for this study is part of the public repositories. GAPSTOP™ is available at https://gitlab.mpcdf.mpg.de/bturo/gapstop_tm[63]. Documentation and installation instructions are

provided for ease of use (https://bturo.pages.mpcdf.de/gapstop_tm). The in silico peak analysis is part of the Contextual Analysis Tools for CryoET and subtomogram averaging (cryoCAT). The source code of cryoCAT is available in the following repository: https://github.com/turonova/cryoCAT[64]. A detailed notebook outlining the parameters and usage of the in silico peak analysis can be found here: https://github.com/turonova/cryoCAT/blob/main/docs/source/tutorials/peak_analysis/peak_analysis.ipynb.

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

## Acknowledgements

This work was funded by the Max Planck Society and the Chan Zuckerberg Initiative for Visual Proteomics Imaging (grant number 2021-234666, M.B., B.T., and G.H.). The Max Planck Computing and Data Facility is acknowledged for computational resources. We thank Stefanie Böhm for critical reading of the manuscript and Sonja Welsch and Iskander Khusainov for fruitful discussions. We also thank Martin Simonovsky for helpful discussions on peak analysis, Jürgen Köfinger, and Jakob Bullerjahn for discussions on the Gaussian mixture model, Agnieszka Obarska-Kosinska for the help with the template of the human NPC subunit, and Huaipeng Xing for the assistance with the human tilt series.

## Author contributions

S.C.L., M.B., B.T., and G.H., conceived the project. P.C.H. and M.W.T. acquired the data. S.C.L. and J.P.K. analyzed the data, and S.L.S. performed the MD simulations. S.C.L., T.M., S.K., and B.T. wrote code. K.G. contributed to the analysis and interpretation of the vault protein results. S.C.L., J.P.K., S.L.S., M.B., B.T., and G.H. wrote the manuscript, and S.C.L., T.M., P.C.H., J.P.K., M.W.T., S.L.S., K.G., B.T., M.B., and G.H. edited the manuscript. M.B., B.T., and G.H. supervised the project and obtained funding.

## Funding

## Competing interests

The authors declare no competing interests.
