## [Peer Review File · Nature Communications]

High-confidence 3D template matching for cryo-electron tomographyEditorial Note: This manuscript has been previously reviewed at another journal that is not operating a transparent peer review scheme. This document only contains reviewer comments and rebuttal letters for versions considered at *Nature Communications*.

REVIEWER COMMENTS

Reviewer #1 (Remarks to the Author):

The manuscript is much improved, and more suitable to publication in Nature Communications.

> I am satisfied with most of the changes implemented in the revised version, in particular I much appreciate the effort to add documentation, tutorials and examples to facilitate usage of the software by the broader community.

> New Figure 2a: what is on the y axis on the right side of the plot?

> Response to my previous comments on Figure 2D, the authors claim that “The correlation between CC and fraction of overlapping voxels is not perfect”, I am still not convinced this is non-trivial. It would be expected that the CC value correlates with the fraction of overlapping voxels, likewise it is expected that this is not a perfectly linear relationship for a number of reasons, including the influence of non-overlapping voxels and the relative intensities of overlapping voxels (assuming overlap can be either 0 or 1). More importantly, the authors also claim that “the dependence on shape gives optimisation potential”. This is an important point but an explanation or practical guidance on how to use the CC curves to optimise template matching. How does one go about changing search parameters given a certain predicted relationship?

> Eye-balling Figure R2b (correlation between z score and defocus) suggests a negative correlation, however the Pearson coefficient indicates a moderate positive correlation. It might be that I am seeing a trend that it is not there? Can the authors explain, or show the trend line? I believe, even if weakened by other complexity factors, a correlation with defocus that seems to indicate higher z scores for closer-to-focus tomogram would be an interesting observation.

> Regarding computational speed, I appreciate the inclusion of some numbers. However the authors still fail to acknowledge the pytom_tm work (PMID: 37686180), which offers similar approaches and is very fast. In addition it is very easy to use and well-documented.

Reviewer #2 (Remarks to the Author):

This article by Cruz-León and colleagues describes the application of template matching to localizing particles and membranes in cryogenic electron tomograms with their latest software version. They show how processing parameters must be optimized for specific targets to account for variations in particle size and shape. Using a GPU implementation of existing CPU methods they make feasible searching with finer angular sampling and improve the overall precision. Using 32 tomograms and multiple templates they walk through the various considerations and show, among other things, the impact of voxel size, masking, filtering, and the angular step size used for searching orientations and identifying targets. More importantly, they provide tools to help users optimize these parameters for specific targets. I did not personally try to use the software tools provided through the links in the article or the rebuttal, but I hope they are continuously refined long after publication and with extensive feedback from the user-base. The paucity of user-friendly software tools to aid in the ever-increasing demands of data analysis makes this work an important contribution that the field of cryo-ET will certainly appreciate. However, there are still a few points I would kindly ask the authors to address before recommending it for publication.

- The new GPU implementation allows for much finer angular sampling than was previously possible. The discussion about the Crowther criterion in the rebuttal raises the question about where the useful upper limit is. The authors convincingly argue that Crowther massively over-estimates the number of rotations needed to reliably detect a particle and that, for example, the NPC can be reliably found with much coarser sampling. However, the only data shown in this direction is Fig 3d. It would be helpful to show how the percentage of particles found for different particles varies with sample angle/number of rotations. I think this would also be needed to move beyond just ribosomes which the authors correctly state is a limitation of existing literature.

- I think it would be useful to calculate the F1 score for all the templates discussed in this manuscript and summarized in a table, along with the molecular weights of each of the templates. Again, if the authors claim their approach goes beyond ribosomes they should actually show this. While the F1 metric is common enough, it would also still be important to include the formula for how the authors calculated it in the materials and methods.

- The evidence for the 100 kDa mass resolution is not convincing as presented. In Fig 3b, middle of bottom row, there are no obvious repetitive peaks in the z-scores for the tubulin dimers. The number of dimers can be estimated from the length the microtubule and number of protofilaments, giving some estimate for the ground truth to estimate the F1 score. Performance will likely vary by protofilament, but it would be helpful to readers that this is quantified and reported. Related, I see no need to repeat these data in Supplemental Fig 8, or the fact that it is repeated should be explicitly stated in the caption. In this caption I would also recommend removing the 6sigma to avoid confusion, and if z-score > 5 was not used as described in Methods, it should be addressed.

- In Fig 2C, the range of mask sizes tested for the optimization are confusing. The images below the plot refer to diameters that are inconsistent with the plot (radius). Also, it would make sense to me that any optimization should search values both smaller and larger than the object (perhaps for some targets excluding flexible exterior regions improves the overall z-score). The plotted range for microtubules does not include a data point for the inner diameter given in the table of Crowther values. How much does the z-score for ribosomes suffer if the mask is too small?
- In Fig 3B there are regions with high z-scores in the top-left and bottom-left regions that are not shown as "Peaks TM" in panel C. How they were excluded is not clear from the materials and methods as currently written. Related, the need to still manually remove obviously incorrect z-score peaks (e.g. atomic membrane model matching with microtubule) should be explicitly stated somewhere to not confuse the novice reader and set realistic expectations.
- In the abstract the authors state the "...confidence 10 to 100-fold above the noise level that outperforms previous methods." However, in the main text there is no obvious place where this claim is substantiated. There is also no discussion about noise levels or a direct comparison to noise in previous methods that I could find. I suggest to include this data or remove the claim from the abstract.
- Text refers to Fig 3F as showing orientations when it shows distances. No reference to Fig 3E in text.
- There is still some inconsistency between sigma and z-score threshold throughout that needs to be addressed, for example in the sections "Detection of subunits and conformer subpopulations" and "High-confidence TM identifies vault encapsulated ribosomes in situ" and where results from an in silico analysis are not presented. If variance of the z-score was used, this is not clear from the Methods as written.
- Fig 5b shows many "TM found" 80S particles that are not also "Annotated", but it is not clear how many of these are false positives. It would be good to perhaps process this subset separately to get an estimate to mention in the text/caption. This information might also be incorporated into Fig 5e, but it is not obvious or easy to me how to pull it out. The caption is also confusing for Fig 5, e.g. there is no pink/blue in panels b & c.
- In the discussion, Vaults are referred to as low-density objects, where I think the authors mean low abundance. Need to also be careful to make clear to the reader where "particle density" refers to number per unit volume and where it refers to how strongly it scatter electrons.
- Pixel size for H. sapiens data are not given in Methods section.
- The runtimes/hardware mentioned in the rebuttal will be of interest to readers and should be included with more details, e.g. "bin4" does not tell you the total number of voxels in the tomogram.
- Maybe it is my misunderstanding of exactly how novaCTF works, but thought that the particle position (height within the tomogram) was required to accurately correct the CTF. From the methods as written, it is not clear how novaCTF was used when the particle position is not known.
- Reference "Fawcett 2004" is not included in bibliography.
- EMPAIR accession code not given in Fig 3 and no placeholder in others. Maybe remove from all?

- Supporting video showing TM results for the nuclear envelope using the three membrane templates appears to have gotten lost in the submission.

In conclusion I see the scientific value of this work and would be eager to see the benefits of improved template matching applied to the increasing amounts of lamella data being produced. However, in my view the manuscript needs a bit more work before it is ready for publication.

Reviewer #3 (Remarks to the Author):

In this revised manuscript from Cruz-Leon and colleagues, the authors make a number of changes to the text and figures as well as share the repository for a python/GPU-accelerated version of the STOPGAP template matching algorithm, which they term GAPSTOPTM. Overall, the authors have satisfactorily addressed all the comments I had for the prior version of the manuscript. I think it reads well and has a much improved presentation of the experimental findings. The GAPSTOPTM repository is still a work in progress, but is beyond the scope of this manuscript, so I don't think this should be considered an impediment to publication. Taken together, I have no further comments and recommend this manuscript for publication.

Reviewer #3 (Remarks on code availability):

I had a brief assessment of the code, and though a work in progress, I think it will be a useful resource to the community. Sufficient documentation is present and the code is functional.

That the code is in progress should not be an impediment for this manuscript, as this code is a performance-optimized version of STOPGAP, which was independently developed by another group and has long been available online.

Point-by-point response to reviewer comments

Note: text in blue indicates quotations from the main manuscript. Figures and tables labeled as Fig. R... and Table R... refer to figures and tables in the rebuttal.

Reviewer #1 (Remarks to the Author):

The manuscript is much improved, and more suitable to publication in Nature Communications.

> I am satisfied with most of the changes implemented in the revised version, in particular I much appreciate the effort to add documentation, tutorials and examples to facilitate usage of the software by the broader community.

Author reply: We thank the reviewer for the positive assessment of our work and the helpful comments and suggestions.

> New Figure 2a: what is on the y axis on the right side of the plot?

Author reply: We thank the reviewer for catching this mistake. We removed the y2-tic marks on the right of Fig. 2a.

> Response to my previous comments on Figure 2D, the authors claim that “The correlation between CC and fraction of overlapping voxels is not perfect”, I am still not convinced this is non-trivial. It would be expected that the CC value correlates with the fraction of overlapping voxels, likewise it is expected that this is not a perfectly linear relationship for a number of reasons, including the influence of non-overlapping voxels and the relative intensities of overlapping voxels (assuming overlap can be either 0 or 1). More importantly, the authors also claim that “the dependence on shape gives optimisation potential”. This is an important point but an explanation or practical guidance on how to use the CC curves to optimise template matching. How does one go about changing search parameters given a certain predicted relationship?

Author reply: In response, we now state that

p.6: “The *in silico* evaluation of multiple templates showed that the CC depends almost linearly across different templates on the fraction of overlapping voxels between the rotated template and the object (Fig. 2d,e), a relation that would be exact if voxel intensities were strictly zero or one.”

We now also provide guidance on how to turn information on the dependence of CC on half-vault orientation (Supplementary Fig. 2) and the NPC subunit (new Supplementary Fig. 4) into a recommended value for the angular sampling, and we then illustrate the effect in the new Supplementary Figs. 8 and 9. We discuss this point in the main text:

p.6: “Therefore, our pipeline allows us to optimize parameters *in silico* in a template specific manner, prior to analyzing experimental data, to then channel the available computational power towards those templates that require more fine-grained scanning. For example, the variation of cross-correlation with angular distance (Supplementary Figs. 2b, 3b, and 4b) provides an initial guide for estimating axis-dependent angular steps. In our experience, angular steps that result in <40% decrease in cross-correlation are considered sufficient, as illustrated for the Vault (Supplementary Figs. 2 and 9)

and the NPC subunit (Supplementary Figs. 4 and 8). Our python based tool will allow users to do this systematically for any template.”

> Eye-balling Figure R2b (correlation between z score and defocus) suggests a negative correlation, however the Pearson coefficient indicates a moderate positive correlation. It might be that I am seeing a trend that it is not there? Can the authors explain, or show the trend line? I believe, even if weakened by other complexity factors, a correlation with defocus that seems to indicate higher z scores for closer-to-focus tomogram would be an interesting observation.

Author reply: The reviewer is correct about the observation in Figure R2b. Upon revision, we acknowledge that there was indeed an error in the labels of R2b (Pearson coefficients) in the correlation plots. We have corrected the labels and included the fitted trend line to provide a clearer representation of the correlation. See the updated Figure R2b below. Note that the data is unchanged and the conclusion remains that we do not see any firm correlations.

Fig. R2: Statistics of z-scores and relations with tomogram thickness and defocus. a, box plots of the z-score for 25 *D. discoideum* tomograms sorted by tomogram thickness (line: median; box: interquartile range; error bars: range). **b,c** Correlation plots of the mean \pm std of the z-scores as a function of the defocus (**b**) and estimated tomogram thickness (**c**). Tomogram thickness was estimated manually as described in Ref. ¹.

> Regarding computational speed, I appreciate the inclusion of some numbers. However the authors still fail to acknowledge the *pytom_tm* work (PMID: 37686180), which offers similar approaches and is very fast. In addition it is very easy to use and well-documented.

Author reply: We thank the reviewer for alerting us of this omission. We now mention pytom_tm (now reference 41 in the manuscript) specifically for its computational speed.

p. 12: “Therefore, we developed and released GAPSTOP™ to harness the GPU's parallel capabilities for TM while preserving the unique missing wedge and noise correlation modeling of the STOPGAP implementation³⁶. Significant speedups by using GPUs were also reported for pytom_tm⁴¹.”

Reviewer #2 (Remarks to the Author):

This article by Cruz-León and colleagues describes the application of template matching to localizing particles and membranes in cryogenic electron tomograms with their latest software version. They show how processing parameters must be optimized for specific targets to account for variations in particle size and shape. Using a GPU implementation of existing CPU methods they make feasible searching with finer angular sampling and improve the overall precision. Using 32 tomograms and multiple templates they walk through the various considerations and show, among other things, the impact of voxel size, masking, filtering, and the angular step size used for searching orientations and identifying targets. More importantly, they provide tools to help users optimize these parameters for specific targets. I did not personally try to use the software tools provided through the links in the article or the rebuttal, but I hope they are continuously refined long after publication and with extensive feedback from the user-base. The paucity of user-friendly software tools to aid in the ever-increasing demands of data analysis makes this work an important contribution that the field of cryo-ET will certainly appreciate. However, there are still a few points I would kindly ask the authors to address before recommending it for publication.

Author reply: We thank the reviewer for the positive assessment of our work and the helpful comments and suggestions, which we address below.

- The new GPU implementation allows for much finer angular sampling than was previously possible. The discussion about the Crowther criterion in the rebuttal raises the question about where the useful upper limit is. The authors convincingly argue that Crowther massively over-estimates the number of rotations needed to reliably detect a particle and that, for example, the NPC can be reliably found with much coarser sampling. However, the only data shown in this direction is Fig 3d. It would be helpful to show how the percentage of particles found for different particles varies with sample angle/number of rotations. I think this would also be needed to move beyond just ribosomes which the authors correctly state is a limitation of existing literature.

Author reply: In response, we have extended our systematic analysis of the dependence on angular sampling from ribosomes (Fig. 3d and Supplementary Fig. 4) and membranes (Fig. 4) to NPC subunits and the Vault, as detailed in the new Supplementary Figs. 7 and 8. These additional calculations provide a more comprehensive understanding of the performance of our approach across different biological structures.

Supplementary Fig. 8: Template matching results for the NPC subunit (C8-symmetric rotational segment) depending on the angular sampling. Cross correlation (z-score) maps obtained on the same tomographic volume (Fig. 1 in the main text) for the NPC subunit (SU) without membrane (*D. discoideum*) using an angular step of 5 (a), 10 (b), 20 (c) and 30 (d) degrees. Labels indicate the number of visible peaks/total number of subunits in the score map.

Supplementary Fig. 9: Template matching results for the half vault depending on the angular sampling. Slice of the cross correlation (z-score) maps obtained on the same tomographic volume (Fig. 1 in the main text) for the half vault (*D. discoideum*) using an angular step of 5 (a), 10 (b), 20 (c) and 30 (d) degrees.

- I think it would be useful to calculate the F1 score for all the templates discussed in this manuscript and summarized in a table, along with the molecular weights of each of the templates. Again, if the authors claim their approach goes beyond ribosomes they should actually show this. While the F1 metric is common enough, it would also still be important to include the formula for how the authors calculated it in the materials and methods.

Author reply: In addition to the ROC curves previously requested by the reviewer, we now also report the F1 score calculations, for ribosomes, NPC, vaults, microtubules, using different search parameters and templates (see the updated Supplementary Figs. 4, 6, 9, 10), which also complements our previous F1 scores for FAS and ribosomes for the *S. Pombe* dataset. We also include the equation used to determine the F1 score in the Methods section (p. 17).

Notably, we now show how the F1 score depends on angle sampling (ribosomes), voxel sampling (ribosomes), template (NPC SU), and template size (microtubules).

We firmly believe that we have provided plenty of evidence that high confidence template matching for CryoET extends well beyond ribosomes.

- The evidence for the 100 kDa mass resolution is not convincing as presented. In Fig 3b, middle of bottom row, there are no obvious repetitive peaks in the z-scores for the tubulin dimers. The number of dimers can be estimated from the length the microtubule and number of protofilaments, giving some estimate for the ground truth to estimate the F1 score. Performance will likely vary by protofilament, but it would be helpful to readers that this is quantified and reported. Related, I see no need to repeat these data in Supplemental Fig 8, or the fact that it is repeated should be explicitly stated in the caption. In this

caption I would also recommend removing the 6 sigma to avoid confusion, and if z-score > 5 was not used as described in Methods, it should be addressed.

Author reply: We believe the reviewer refers to Figure 6b, not 3b. In response, we now plot the z-scores along the axial lines passing through protofilaments 12 and 8 as labeled in Fig 6b (Supplementary Fig. 11e,f). In the longitudinal profiles of the z-scores it is evident, for both the protofilament template (red) and the tubulin dimer (green), that the local maxima correspond to the arrangement of the lattice.

Supplementary Fig. 12: Statistics for template matching using a single ~100 kDa $\alpha\beta$ -tubulin dimer. a,b, Cross correlation (z-score) maps obtained on the same tomographic volume using a template and a cylindrical mask (top view) containing ~4 microtubule protofilaments (three and two halves) (a) and a single $\alpha\beta$ -tubulin dimer (b). Note that the templates and scores maps on a and b are identical to Fig. 6b in the main text. c, Positions of the individual subunits extracted from (a), and adopted as annotated

particles. Particles colored in brown and blue correspond to protofilaments 12 and 8 as labeled in Fig 6b used for analysis in panels e and f. d, Statistical analysis of the $\alpha\beta$ -tubulin dimer localizations in the volume shown in b with the positions in c as reference. Shown are the recall = $\#(\text{true positives}) / [\#(\text{true positives}) + \#(\text{false negatives})]$ (blue, left axis) and the precision = $\#(\text{true positives}) / [\#(\text{true positives}) + \#(\text{false positives})]$ (orange, right axis) as functions of the threshold applied to the tubulin dimer z-score. A position was considered as true positive if it was within 3 nm of a position in (c). e,f, z-scores along the axial lines passing through protofilament 12 for the protofilament template (a) and the $\alpha\beta$ -tubulin dimer template (b). The black circles and vertical lines indicate the positions of the extracted peaks in panel c.

We note this in the main text:

p.10: “Despite the low combined mass of only 100 kDa, TM achieves good statistics both in terms of true positives and (likely) false negatives. Although the subunit segmentation along the filament was blurred, it is evident in the longitudinal z-scores along the axial lines passing through protofilaments that the local maxima correspond to the subunits in the microtubule lattice for both the protofilament template and the $\alpha\beta$ -tubulin dimer (Supplementary Fig. 11).”

Furthermore, as suggested, we explicitly state in the caption of Supplementary Fig. 11 that the score maps and templates from a and b are identical to the panel 6b in the main text. We decided to keep these maps in the Supporting Fig. 11 for clarity.

- In Fig 2C, the range of mask sizes tested for the optimization are confusing. The images below the plot refer to diameters that are inconsistent with the plot (radius). Also, it would make sense to me that any optimization should search values both smaller and larger than the object (perhaps for some targets excluding flexible exterior regions improves the overall z-score). The plotted range for microtubules does not include a data point for the inner diameter given in the table of Crowther values. How much does the z-score for ribosomes suffer if the mask is too small?

Author reply: We thank the reviewer for this comment. To avoid confusion, we make the labels on the images of the masks to refer also to the radius, as in Fig. 2c. Furthermore, we perform new calculations with even smaller masks to show how the z-scores decrease with the radius of the mask. See the updated Fig 2c.

- In Fig 3B there are regions with high z-scores in the top-left and bottom-left regions that are not shown as “Peaks TM” in panel C. How they were excluded is not clear from the materials and methods as currently written. Related, the need to still manually remove obviously incorrect z-score peaks (e.g. atomic membrane model matching with microtubule) should be explicitly stated somewhere to not confuse the novice reader and set realistic expectations.

Author reply: We thank the reviewer for this comment. In fact, the regions in the bottom-left, bottom-right and upper-left corners of the Fig. 3b show peaks that are not included because those parts are outside of the lamellae borders (Fig. 3a). To avoid possible confusion and set realistic expectations, we state this explicitly now:

p.7: “... and selected TM peaks corresponding to local maxima in the z-score map that are above a threshold (Fig. 3) and clearly inside the lamella borders.”

Regarding, the membranes and microtubules, we already state that:

p.8: “False positives for the small templates (atomistic, small STA), e.g., from a microtubule segment (Fig. 4 left; see also Fig. 1) are suppressed by using the large STA template (or, visually, by recognizing the lacking 2D extension).”

Furthermore, as we pointed out in the previous reply letter (see reviewer 3, request 6), “an *a priori* determination of the threshold is still challenging in the current pipeline. We recommend that users visualize the score maps and determine thresholds empirically”.

- *In the abstract the authors state the “...confidence 10 to 100-fold above the noise level that outperforms previous methods.” However, in the main text there is no obvious place where this claim is substantiated. There is also no discussion about noise levels or a direct comparison to noise in previous methods that I could find. I suggest to include this data or remove the claim from the abstract.*

Author reply: We shortened the sentence in the abstract to:

p.1: “...confidence 10 to 100-fold above the noise level”.

- *Text refers to Fig 3F as showing orientations when it shows distances. No reference to Fig 3E in text.*

Author reply: The reviewer is correct. We have corrected the references as follow:

p. 7: “With orientations separated by ~5 degrees, TM detected ~95% of the 437 previously annotated particles with a mean distance to the TM peak of (3.73 ± 1.57) nm (Fig. 3f) and with orientations that closely matched the annotated orientations (Fig. 3e).”

- *There is still some inconsistency between sigma and z-score threshold throughout that needs to be addressed, for example in the sections “Detection of subunits and conformer subpopulations” and “High-confidence TM identifies vault encapsulated ribosomes in situ” and where results from an in silico analysis are not presented. If variance of the z-score was used, this is not clear from the Methods as written.*

Author reply: We thank the reviewer raising this point. We have revised the entire manuscript and supporting material to remove the inconsistencies. To clarify the reviewer question, we did not use the variance of the z-score.

- *Fig 5b shows many “TM found” 80S particles that are not also “Annotated”, but it is not clear how many of these are false positives. It would be good to perhaps process this subset separately to get an estimate to mention in the text/caption. This information might also be incorporated into Fig 5e, but it is not obvious or easy to me how to pull it out. The caption is also confusing for Fig 5, e.g. there is no pink/blue in panels b & c.*

Author reply: In response to your recommendation, we now performed STA separately on the subset of particles from Fig. 5b-d (see Supporting Fig. 9). To minimize potential template bias, we used the peaks obtained from TM using the 60S as template (Supporting Fig. 9g).

To determine the quality of the “TM found” particles, we performed two separate calculations: (i) an average using all peaks obtained from TM (90 particles, see Supporting Fig. 9c,h), and (ii) an average using only the peaks obtained from TM that were not annotated (34 particles, Supporting Fig. 9b,d). In both

cases, we recovered features from the 80S (see Supporting Fig. 9e), demonstrating that the unannotated subset also contains true positives.

Furthermore, we colocalized the peaks obtained from the 40S. A total of 15/34 peaks from the 40S colocalize with the unannotated 60S peaks (Supplementary Fig. 10e). Finally, we complement our STA with an expert inspection of the volume. Upon closer inspection, only a small fraction (4/90) of the peaks do not look like ribosomes, further supporting the robustness of our approach.

The previous analysis shows that TM identifies high quality particles, reporting significantly more particles than Relion and a low false-positive rate. However, determining the individual true positives among the TM peaks requires an analysis with many more particles beyond the scope of this work.

We included these results in the main text and the new Supplementary Fig. 10:

p.9: "Finally, using all the TM peaks detected by the 60S (90 particles) or only the unannotated (34 particles) TM peaks detected by the 60S, we recovered features from the 80S (see Supplementary Fig. 10), demonstrating that the unannotated subset also contains true positives."

Supplementary Fig. 10: Template matching locates high quality ribosomes. **a**, Tomogram volume from Fig. 5a, with TM found particles using the 60S template colored as annotated from Relion (green) and unannotated (blue). Structures (**b,c**) and slices (**d,h**) obtained by averaging the particles from TM using the large ribosomal subunit 60S (**g**) as template using only the unannotated particles (**b, d**) and all the TM particles (**c,h**). Note that in both cases, we recover features of the 80S ribosome (**f**). **e**, Tomogram volume as in **a**, for clarity only the unannotated 60S particles from **a**, and the colocalized peaks using the 40S as template (pink) are shown.

- In the discussion, Vaults are referred to as low-density objects, where I think the authors mean low abundance. Need to also be careful to make clear to the reader where “particle density” refers to number per unit volume and where it refers to how strongly it scatter electrons.

Author reply: We rewrote the sentence as:

p. 11: “Vaults, for example, are of low abundance with a low-contrast interior, but their unique shape facilitates identification with confidence (Fig. 1 and Supplementary Fig. 2)”.

- Pixel size for *H. sapiens* data are not given in Methods section.

Author reply: We correct this omission in the methods as follows:

p. 13: “Projections had a pixel size of 2.176 Å and 1.223 Å for *D. discoideum*, 3.37 Å for *S. pombe*, and 1.223 Å for *H. Sapiens* respectively, and were acquired in a dose symmetric acquisition scheme⁴⁹ with 2 deg increments^{4,50}.”

- The runtimes/hardware mentioned in the rebuttal will be of interest to readers and should be included with more details, e.g. “bin4” does not tell you the total number of voxels in the tomogram.

Author reply: We thank the reviewer for this comment. The number of voxels in the tomogram and template is indeed informative. Here is the updated response:

With STOPGAP, the TM of the 80S template (52x52x52 voxels) with a bin4 tomogram (1024x1024x450 voxels) and angular steps of 10 degrees, running on ~100 CPUs, takes ~3 days. GAPSTOPTM takes ~17 minutes on the same system using Nvidia A100 GPUs. We re-iterate that details of GAPSTOPTM implementations as well as its comprehensive benchmarking are beyond the scope of our current manuscript and will be part of another publication in the near future.

- Maybe it is my misunderstanding of exactly how novaCTF works, but thought that the particle position (height within the tomogram) was required to accurately correct the CTF. From the methods as written, it is not clear how novaCTF was used when the particle position is not known.

Author reply: As described in the original paper², the novaCTF approach performs 3D CTF corrections on the entire tomogram and is completely blind to any content. The defocus value that is estimated for the zero-tilt image is used at the z-center of the tomogram and the defocus value is gradually changed in both directions. Thus, the whole tomogram is corrected, not just selected particles as in other approaches (Warp, or Relion 4).

- Reference “Fawcett 2004” is not included in bibliography.

Author reply: Thank you for pointing out the missing reference. We include the reference to “Fawcett 2004” in the bibliography section of the manuscript, now reference 61.

- EMPAIR accession code not given in Fig 3 and no placeholder in others. Maybe remove from all?

Author reply: Following the reviewer's advice, we remove the EMPAIR access code from the figures. All EMPAIR codes are listed in the Methods section.

- Supporting video showing TM results for the nuclear envelope using the three membrane templates appears to have gotten lost in the submission.

Author reply: We attach again the supporting video showing the template matching results for the nuclear envelope. Note that it corresponds to the case of the large STA template from Fig. 4.

In conclusion I see the scientific value of this work and would be eager to see the benefits of improved template matching applied to the increasing amounts of lamella data being produced. However, in my view the manuscript needs a bit more work before it is ready for publication.

Author reply: We once more thank the reviewer for the positive assessment and the helpful comments and suggestions.

Reviewer #3 (Remarks to the Author):

In this revised manuscript from Cruz-Leon and colleagues, the authors make a number of changes to the text and figures as well as share the repository for a python/GPU-accelerated version of the STOPGAP template matching algorithm, which they term GAPSTOPTM. Overall, the authors have satisfactorily addressed all the comments I had for the prior version of the manuscript. I think it reads well and has a much improved presentation of the experimental findings. The GAPSTOPTM repository is still a work in progress, but is beyond the scope of this manuscript, so I don't think this should be considered an impediment to publication. Taken together, I have no further comments and recommend this manuscript for publication.

Author reply: We thank the reviewer for the positive assessment of our work.

Reviewer #3 (Remarks on code availability):

I had a brief assessment of the code, and though a work in progress, I think it will be a useful resource to the community. Sufficient documentation is present and the code is functional.

Author reply: We thank the reviewer for testing and positively assessing our code.

That the code is in progress should not be an impediment for this manuscript, as this code is a performance-optimized version of STOPGAP, which was independently developed by another group and has long been available online.

References

1. Tuijtel, M. W. et al. *Thinner Is Not Always Better: Optimising Cryo Lamellae for Subtomogram*

Averaging. <http://biorxiv.org/lookup/doi/10.1101/2023.07.31.551274> (2023)

doi:10.1101/2023.07.31.551274.

2. Turoňová, B., Schur, F. K. M., Wan, W. & Briggs, J. A. G. Efficient 3D-CTF correction for cryo-electron tomography using NovaCTF improves subtomogram averaging resolution to 3.4 Å. *J. Struct. Biol.* **199**, 187–195 (2017).

REVIEWERS' COMMENTS

Reviewer #1 (Remarks to the Author):

I am satisfied with all responses to my comments, and recommend publication in Nature Communications.

Reviewer #2 (Remarks to the Author):

I thank the authors for the effort in including additional data and for thoroughly addressing my comments. I am convinced by all the replies and my questions have been answered, much appreciated!

Two very minor things were noticed:

Line 152: "...clear guidelines ON how to adjust..."

Line 335: inconsistent omission of "F1" and inclusion of "=" between the parentheses